# Presence versus absence of *CYP734A50* underlies the style-length dimorphism in primroses

Cuong Nguyen Huu[1], Christian Kappel[1], Barbara Keller[2], Adrien Sicard[1], Yumiko Takebayashi[3], Holger Breuninger[4], Michael D Nowak[2,5], Isabel Bäurle[1], Axel Himmelbach[6], Michael Burkart[7], Thomas Ebbing-Lohaus[8], Hitoshi Sakakibara[3], Lothar Altschmied[6], Elena Conti[2], Michael Lenhard[1]*

[1]Institute for Biochemistry and Biology, University of Potsdam, Potsdam-Golm, Germany; [2]Department of Systematic and Evolutionary Botany, University of Zurich, Zurich, Switzerland; [3]RIKEN Center for Sustainable Resource Science, Yokohama, Japan; [4]Department of Plant Science, University of Oxford, Oxford, United Kingdom; [5]Natural History Museum, University of Oslo, Oslo, Norway; [6]Leibniz Institute of Plant Genetics and Crop Plant Research, Gatersleben, Germany; [7]Botanical Garden, University of Potsdam, Potsdam, Germany; [8]Gartenbau Ebbing-Lohaus, Heiden, Germany

**Abstract** Heterostyly is a wide-spread floral adaptation to promote outbreeding, yet its genetic basis and evolutionary origin remain poorly understood. In *Primula* (primroses), heterostyly is controlled by the *S*-locus supergene that determines the reciprocal arrangement of reproductive organs and incompatibility between the two morphs. However, the identities of the component genes remain unknown. Here, we identify the *Primula CYP734A50* gene, encoding a putative brassinosteroid-degrading enzyme, as the *G* locus that determines the style-length dimorphism. *CYP734A50* is only present on the short-styled S-morph haplotype, it is specifically expressed in S-morph styles, and its loss or inactivation leads to long styles. The gene arose by a duplication specific to the Primulaceae lineage and shows an accelerated rate of molecular evolution. Thus, our results provide a mechanistic explanation for the *Primula* style-length dimorphism and begin to shed light on the evolution of the *S*-locus as a prime model for a complex plant supergene.

*For correspondence: michael. lenhard@uni-potsdam.de

## Introduction

Heterostyly is a prominent, wide-spread adaptation to promote outbreeding (*Barrett, 2002*, *1992*). Individuals of heterostylous species belong to two or three morphs with reciprocal reproductive-organ arrangement and intra-morph incompatibility (*Barrett, 2002*). Since Darwin's seminal book on *The different forms of flowers on plants of the same species* (*Darwin, 1877*), *Primula* (primroses) has been the prime model for studying heterostyly (*Barrett, 1992*; *Gilmartin, 2015*; *Weller, 2009*). Heterostyly in *Primula* is controlled by the *S*-locus supergene that likely comprises five tightly linked genes, determining style length (the so-called *G* locus), anther height, pollen size and male and female intra-morph incompatibility (*Figure 1A*; *Figure 1—figure supplement 1*) (*Ernst, 1936*; *Lewis and Jones, 1992*). The S-morph haplotype carries dominant alleles at all loci, resulting in short styles due to reduced cell elongation (*Webster and Gilmartin, 2006*) and anthers at the mouth of the flower; the L-morph haplotype carries the recessive alleles, causing long styles and low anthers. S-morphs are heterozygous for the two haplotypes, and L-morphs homozygous

**eLife digest** Flowers are highly specialized structures that many plants use to reproduce. Male organs called stamens on the flowers make pollen that can be transferred – usually by insect carriers or the wind – to a female structure called the stigma on another plant. However, since many flowers contain both male and female organs, it is also possible for the pollen to land on the stigma of the same flower, leading to a process called "self-fertilization".

Many plants have developed mechanisms that prevent self-fertilization. For example, primroses produce two different types of flowers that arrange their stamens and stigmas differently. The stigma sits on the top of a stalk known as the style. Some primroses produce flowers with short stamens and a long style, resulting in the stigma being located high up in the flower ("pin" flowers), while others produce flowers with a short style and long stamens ("thrum" flowers). Primrose pollen is carried by insects and the different lengths of the styles and stamens make it more likely that pollen from a pin flower will land on the stigma of a thrum flower instead of a pin flower (and vice versa).

Although primrose flowers have fascinated botanists for centuries, the genes responsible for making the two types of flower had not been identified. Genetic studies indicated that different genes control the length of the stamens and style. However, these genes appear to be very close to each other on primrose DNA, which made it difficult to study them individually.

Huu et al. identified a gene called *CYP734A50* that is responsible for the difference in style length in the flowers of a primrose called *Primula veris*. The gene is only present in the plants that have thrum flowers across a wide range of primrose species and genetic mutations that inactivate the gene in these plants result in flowers with longer styles. *CYP734A50* encodes an enzyme that breaks down plant hormones called brassinosteroids, which normally promote growth. Treating thrum flowers with brassinosteroids increased the length of the styles. Future challenges are to identify the other genes that are responsible for producing pin and thrum flowers and to understand how this group of genes evolved.

recessive; the absence of homozygotes for the S-morph haplotype likely results from a recessive lethal mutation on this haplotype (*Kurian and Richards, 1997*). Efficient fertilization is only possible in inter-morph crosses. A candidate for a causal distyly gene has been described from *Fagopyrum* (*Yasui et al., 2012*); also, S-locus linked sequences have been identified from *Primula* and *Turnera* (*Labonne and Shore, 2011*; *Labonne et al., 2009*; *Li et al., 2007*, *2015*; *Nowak et al., 2015*). However, in *Primula* no causal distyly gene has been isolated, and no plausible mechanistic explanation for style length dimorphism is available in any species, hampering progress in understanding the evolution of the S-locus supergene.

## Results and discussion

We performed transcriptome sequencing on styles versus corolla tubes including fused stamens from *P. veris* L- and S-morphs. Candidates for the *G* locus were selected based on stronger expression in S-morph styles than S-morph corollas and both organs in L-morphs; this identified 11 genes, seven of which were almost exclusively expressed in S-morphs (*Figure 1—source data 1*). We focussed on transcript DN148700_c0_g1, encoding CYP734A50 as a member of the cytochrome P450 CYP734A-family that degrade brassinosteroids (*Ohnishi et al., 2006*; *Thornton et al., 2011*), a class of plant hormones promoting cell elongation. No reads mapping to *CYP734A50* were detected in L-morph samples (*Figure 1B*; *Figure 1—source data 1*). RT-PCR confirmed specific expression in styles of S-morphs of *P. veris* and the closely related *P. vulgaris* (*Figure 1C*; *Figure 1—figure supplement 2*; *Figure 1—figure supplement 3*). Genotyping *P. veris* and *P. vulgaris* individuals indicated that the gene was present in each of 154 *P. veris* S-morphs from a natural population exhibiting very short linkage disequilibrium close to the S-locus (*Nowak et al., 2015*), while none of the five exons could be amplified in any of 151 L-morph plants; the same was seen in 41 S- and 40 L-morph plants of *P. vulgaris* (*Figure 2A,B*; *Figure 2—figure supplement 1*). A public genome-sequencing dataset of *P. vulgaris* contained essentially no reads mapping to *CYP734A50* from

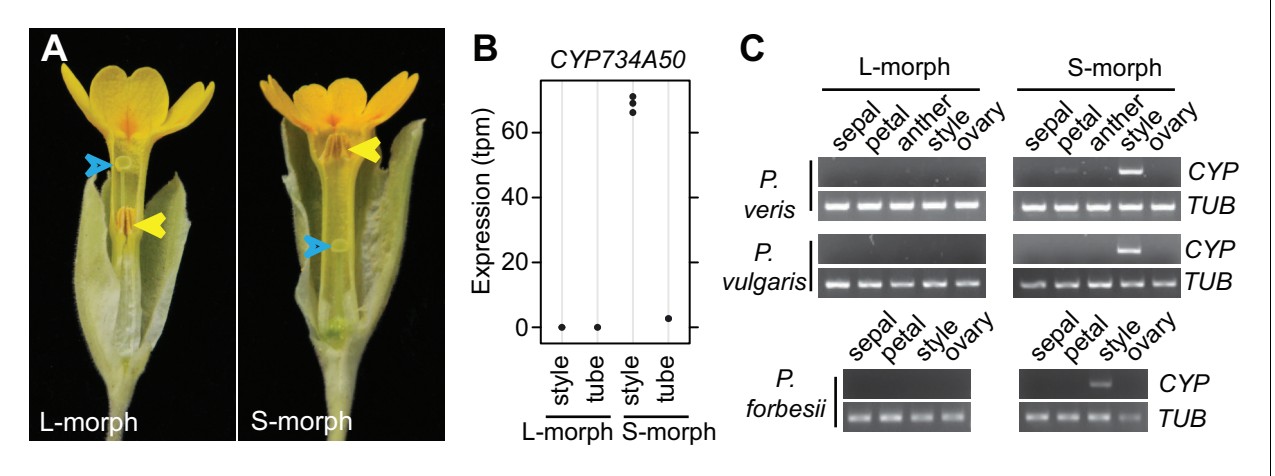

**Figure 1.** Identification of *CYP734A50* as an S-morph and style-specific gene. (**A**) Dissected *P. veris* L- (left) and S-morph flowers (right), showing reciprocal differences in style length and anther position (yellow arrowheads). Blue arrowheads indicate stigmas on top of styles. (**B**) Expression levels of *CYP734A50* in *P. veris* based on RNA-seq (*n* = 3 biological replicates each). See also *Figure 1—source data 1*. (**C**) Expression of *CYP734A50* and *TUBULIN* in dissected floral organs. Full gel images and qRT-PCR quantification based on three biological replicates are shown in *Figure 1—figure supplement 2* and *3*.

The following source data and figure supplements are available for figure 1:

**Source data 1.** Candidates for S-morph style-specific genes.
**Figure supplement 1.** Schematic of the genetic control of heterostyly in *Primula*.
**Figure supplement 2.** Style-specific expression of *CYP734A50*.
**Figure supplement 3.** Quantification of style-specific expression of *CYP734A50*.

L-morph samples; in S-morph samples, the read coverage of *CYP734A50* exons was only half as deep as for a related *CYP* gene (see below) (*Figure 2—source data 1*). Together, these observations indicate that the hemizygous *CYP734A50* locus is only present on the S-morph haplotype in strict linkage to the dominant *G* allele.

Heterostyly evolved only once in the genus *Primula* (*Mast et al., 2006*). Thus, the same causal genes should co-segregate with morph phenotypes in all heterostylous *Primula* species. To test this for *CYP734A50*, we sequenced style transcriptomes of the distantly related *P. forbesii* S- and L-morphs (divergence time *P. veris* – *P. forbesii* ~20–25 Mya [*de Vos et al., 2014*]). Again, the *P. forbesii CYP734A50* homologue was exclusively expressed in styles of S-, but not L-morphs (*Figure 1C*; *Figure 1—source data 1*; *Figure 1—figure supplement 2*). Each of its five exons was PCR-amplified from 10 *P. forbesii* S-morphs, but none of 10 L-morphs (*Figure 2C*; *Figure 2—figure supplement 1*). Genome sequencing of pooled S- versus L-morphs detected no *CYP734A50*-specific reads in L-morphs, and read coverage in S-morphs was only half that of a closely related gene (*Figure 2—source data 1*). Thus, the presence/absence polymorphism of *CYP734A50* has been maintained as a trans-specific polymorphism in strict linkage with the *G* locus over more than 20 million years of evolution, indicating that it is under strong balancing selection, as predicted for a causal heterostyly locus (*Mast and Conti, 2006*).

The hypothesis that *CYP734A50* represents the *G* locus and limits cell expansion by degrading brassinosteroids makes three testable predictions. First, loss of *CYP734A50* activity should lengthen styles; second, S-morph styles should contain lower amounts of active brassinosteroids than L-morph styles; third, cell expansion of S-morph styles should be rescued by brassinosteroid treatment.

We assayed the *CYP734A50* genotype in seven derived long homostyles with independent evolutionary origins (*Mast et al., 2006*). Classic models predict that long homostyles originate via

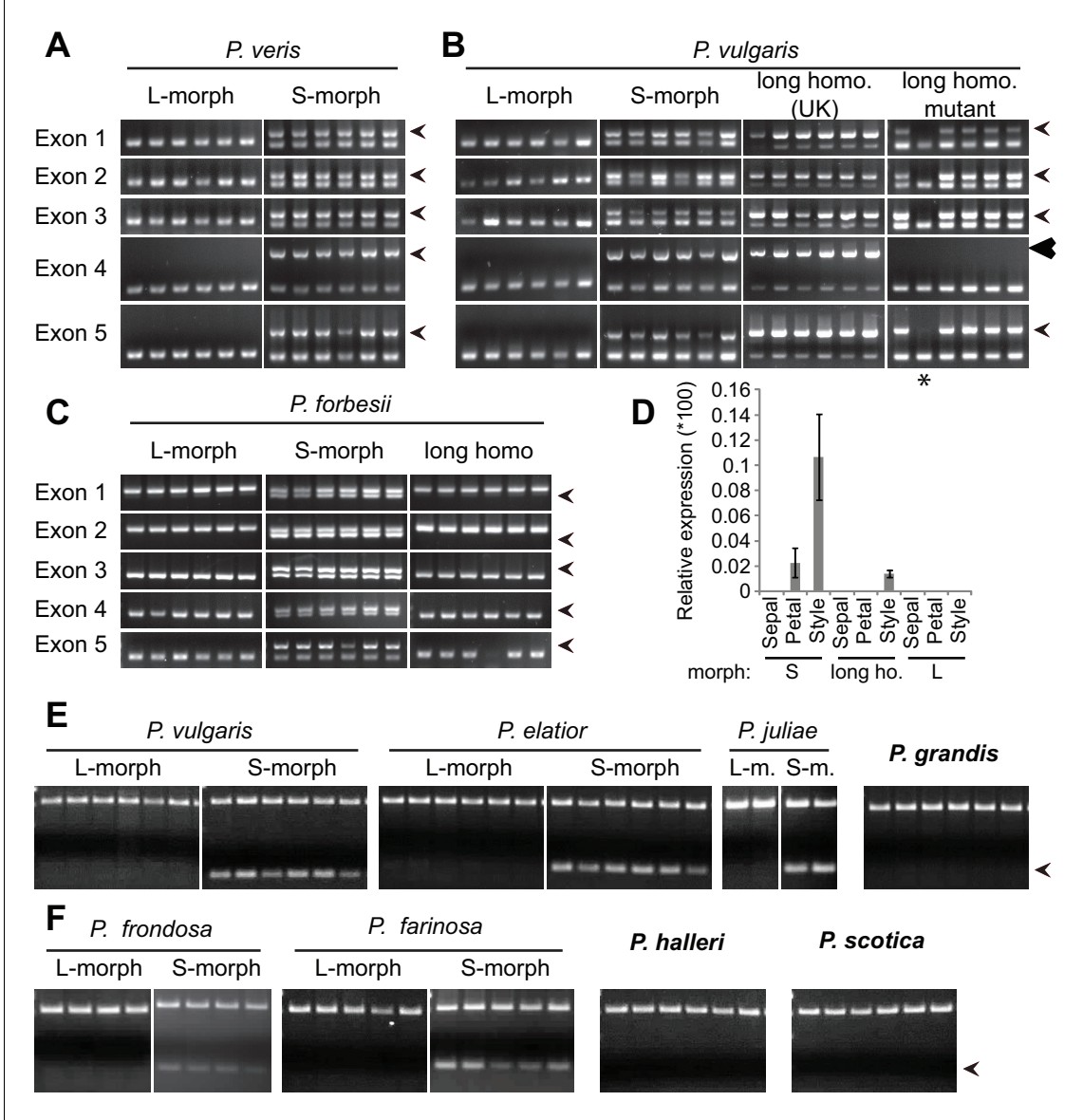

**Figure 2.** Co-segregation of *CYP734A50* with the style phenotype. (A–C) PCR-genotyping of individuals with the indicated phenotypes of *P. veris* (A), *P. vulgaris* (B) and *P. forbesii* (C). 'Long homo (UK)' are naturally occurring long homostyles from England; 'long homo mutant' are long homostyles from the commercially grown population. 'Long homo' *P. forbesii* are long-homostylous putative recombinants found in our experimental population. Multiplex PCR amplified *CYP734A50* exons (arrowheads) and *ITS* as an internal control. Enlarged arrowhead marks absence of exon 4 in the mutant. Asterisk in (B) marks a segregating L-morph individual. (D) Expression of *CYP734A50* in dissected floral organs of naturally occurring long homostyles, S- and L-morphs of *P. vulgaris*, normalized to α-*TUBULIN*. Values are mean ± SD from three biological replicates. (E,F) PCR-genotyping of individuals with the indicated phenotypes from species in *Primula* sect. *Primula* (E) and *Primula* sect. *Aleuritia* (F). Long homostylous species are indicated in bold. Multiplex PCR was performed using primers to amplify exon 3 of *CYP734A50* (arrowhead) and *ITS* as an internal control. Full gel images are shown in *Figure 2—figure supplement 2*. Specificity of PCR amplification for *CYP734A50* is shown in *Figure 2—figure supplement 3*. Each set of plants was genotyped at least twice independently.

The following source data and figure supplements are available for figure 2:

**Source data 1.** Read counts from Illumina whole-genome sequencing of S- and L-morph plants of *P. vulgaris* and *P. forbesii*.

**Figure supplement 1.** Co-segregation of *CYP734A50* with the style phenotype.

**Figure supplement 2.** Loss of CYP734A50 from natural long homostylous species.

*Figure 2 continued on next page*

*Figure 2 continued*

**Figure supplement 3.** Specificity of PCR-genotyping in different *Primula* species.
**Figure supplement 4.** Naturally occurring long homostyles of *P. vulgaris*.
**Figure supplement 5.** Characterization of long-homostylous mutants of *P. vulgaris* and *P. x pruhoniciana*.
**Figure supplement 6.** Virus-induced gene silencing in *P. forbesii*.

recombination in the *S* locus affecting both the *G* and the linked female-compatibility locus, hence they are self-compatible (*Barrett, 1992*; *Lewis and Jones, 1992*); by contrast, in self-incompatible long homostyles only the *G* locus should be affected by mutation or recombination (*Figure 1—figure supplement 1*). Indeed, in *Primula* sect. *Aleuritia* (*Mast et al., 2001*; *Richards, 2003*), *CYP734A50* was present in all tested S-, but none of the L-morphs of the heterostylous *P. frondosa* (*n* = 5 each) and *P. farinosa* (*n* = 5 each), and in none of the long-homostylous *P. scotica* (*n* = 8) and *P. halleri* (*n* = 9) (*Figure 2F*; *Figure 2—figure supplement 2*; *Figure 2—figure supplement 3*). Similarly in *Primula* sect. *Primula*, *CYP734A50* was present in all S-, but none of the L-morphs of the heterostylous *P. elatior* (*n* = 15 each) and *P. juliae* (*n* = 2 each), nor in the homostylous *P. grandis* (*n* = 11) (*Figure 2E*; *Figure 2—figure supplement 2*; *Figure 2—figure supplement 3*). Our *P. forbesii* population included self-compatible long homostyles; as expected, none of these (*n* = 6) contained the *CYP734A50* locus (*Figure 2C*; *Figure 2—figure supplement 1*). Natural *P. vulgaris* populations including long-homostyles occur in two regions in southern England (*Crosby, 1940*, *1949*) (*Figure 2—figure supplement 4*). Unexpectedly, the *CYP734A50* locus was present in all tested long homostyles and in the S-morphs, but in none of the L-morphs from both regions (*Figure 2B*). However, quantitative RT-PCR on RNA from styles of the three morphs indicated strongly reduced *CYP734A50* expression in the long homostyles compared to S-morphs (*Figure 2D*). This suggests that a promoter mutation or epigenetic silencing of *CYP734A50* is responsible for the long-style phenotype, providing an alternative route to the formation of self-compatible long homostyles in nature (*Figure 1—figure supplement 1*). We newly identified long-homostylous, yet apparently self-incompatible mutants from commercially grown *P. vulgaris* and *P. x pruhoniciana* (*Figure 3B,C*; *Figure 2—figure supplement 5*). Exon 4 of *CYP734A50* was deleted in both, and no full-length transcript could be detected in the *P. vulgaris* mutants (*Figure 2B*; *Figure 2—figure supplement 5*). Lastly, we used virus-induced gene silencing (VIGS) to downregulate *CYP734A50* expression in *P. forbesii* (*Figure 2—figure supplement 6*). Two treated S-morph individuals showed reduced *CYP734A50* expression and formed longer styles than mock-treated or virus-treated S-morphs with unaffected *CYP734A50* expression (*Figure 2—figure supplement 6*). When the *P. veris* gene was overexpressed in tobacco, leaf and stem growth were not reduced as seen for other *CYP734A* family members (*Ohnishi et al., 2006*; *Thornton et al., 2011*), likely reflecting the considerable sequence divergence of the *P. veris* CYP734A50 protein (see below); this mirrors the failure of certain mouse *SRY* alleles to induce correct male development in a different genetic background (*Albrecht et al., 2003*). Thus, the *CYP734A50* presence/absence polymorphism strictly cosegregates with the morph type in all heterostylous species sampled across the *Primula* phylogeny, and the gene is absent or inactivated in all seven cases of derived long homostyly. Together with the VIGS results, this strongly supports the conclusion that *CYP734A50* is the *G* locus of the *Primula* heterostyly supergene that determines the style-length polymorphism.

If CYP734A50 degrades brassinosteroids, S-morph styles should contain less active brassinosteroids than L-morph styles. While castasterone was present in L-morph styles, it was virtually undetectable in S-morph styles (*Figure 3A*); brassinolide was undetectable in styles of either morph. When treating developing *P. vulgaris* and *P. forbesii* flowers with brassinolide, treated S-morphs developed significantly longer styles in a concentration-dependent manner; by contrast, styles of L-morph plants did not show a consistent response, as did corolla-tubes in either morph, indicating that exogenous brassinolide treatment specifically enhances growth of S-morph styles (*Figure 3B,C*; *Figure 3—figure supplement 1*). This effect was entirely due to increased cell elongation after brassinolide treatment (*Figure 3D*; *Figure 3—figure supplement 1*).

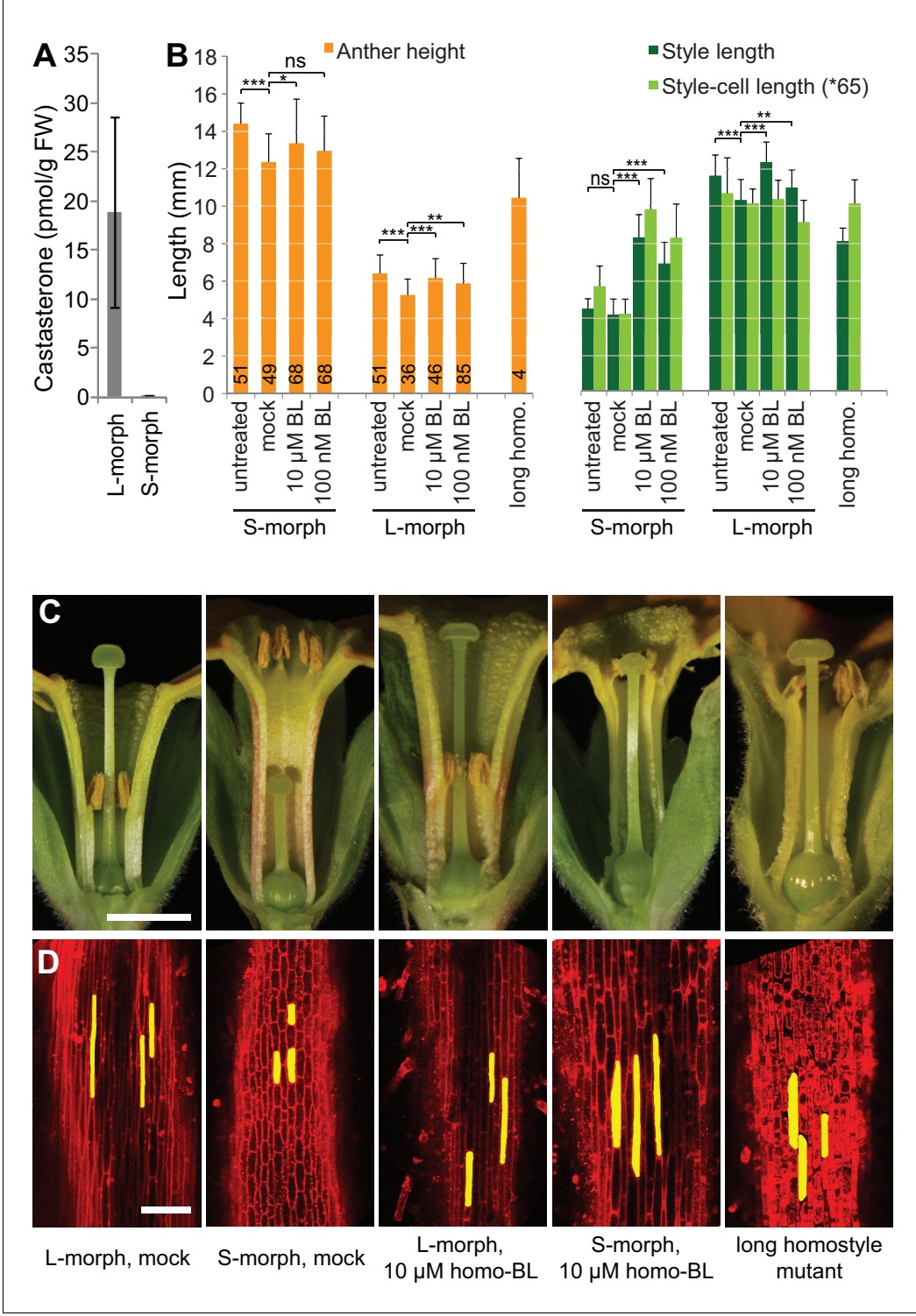

**Figure 3.** Endogenous brassinosteroid content in styles and response to exogenous brassinosteroid treatment in *P. vulgaris*. (**A**) Castasterone levels in styles of *P. vulgaris*. Values are mean ± SD from four biological replicates. (**B**) Anther height, style and style-cell lengths of plants with the indicated phenotypes after brassinolide (BL) treatment. Numbers of flowers used for stamen and style-length measurements are indicated in the bars. Six cells from 10 styles each were measured, except for long homostyle mutant (four styles). Values are mean ± SD. Samples were compared using a two-sided t-test assuming unequal variance, followed by Bonferroni correction. *p<0.05; **p<0.01; ***p<0.001. (**C,D**) Flower phenotypes (**C**) and style cells (**D**) from the indicated treatments.
*Figure 3 continued on next page*

*Figure 3 continued*

Representative cells are marked in yellow. Scale bars: 5 mm (**C**), 100 μm (**D**). A replicate experiment using *P. forbesii* is shown in *Figure 3—figure supplement 1*.

The following figure supplement is available for figure 3:

**Figure supplement 1.** Response of *P. forbesii* flowers to exogenous brassinosteroid treatment.

*P. veris* and *P. forbesii* genome and transcriptome sequences identified a closely related *CYP734A50* paralogue, *CYP734A51*, in both morphs (*Figure 2—source data 1*). A molecular phylogeny of angiosperm CYP734A proteins found in sequenced genomes indicated that *CYP734A50* arose via a gene duplication specific to the *Primula* lineage (*Figure 4*; *Figure 4—figure supplement 1*). Like the male sex-determining gene *SRY* in mammals (*O'Neill and O'Neill, 1999*; *Tucker and Lundrigan, 1995*), a causal heterostyly gene present on only one of the two homologous chromosomes in a non-recombining region should be subject to two opposing effects. The effective population size for such a gene is approximately one fourth that of unlinked diploid genes, resulting in reduced efficacy of selection and stronger drift, including the accumulation of repetitive elements

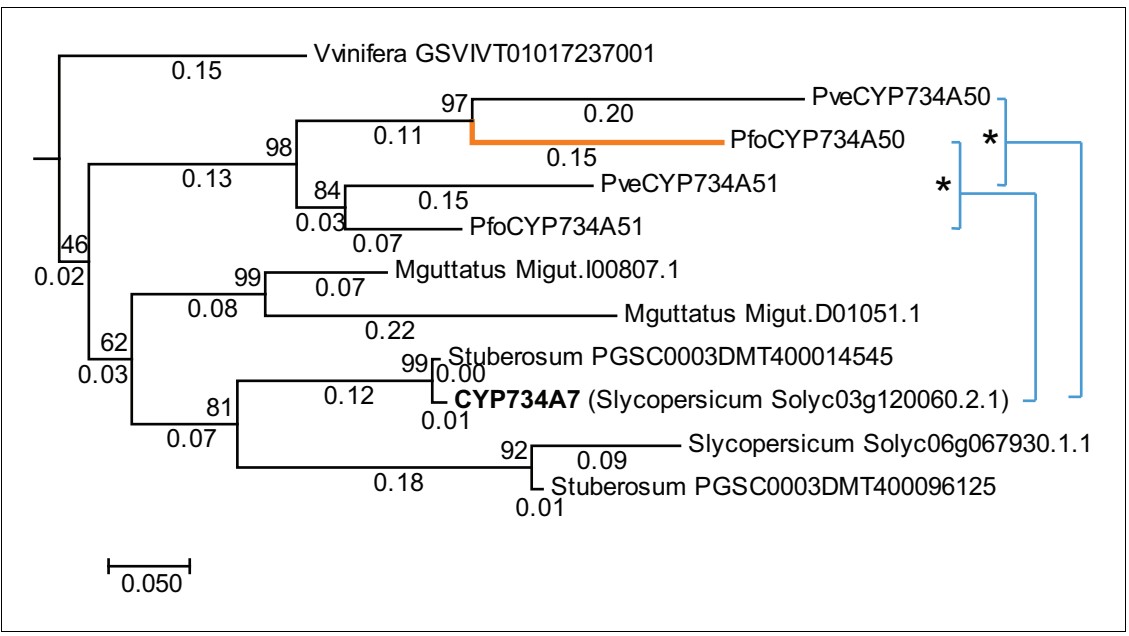

**Figure 4.** Phylogenetic analysis of CYP734A-like proteins. Molecular phylogeny of CYP734A50-related proteins based on Neighbour Joining. The tree is drawn to scale, with branch lengths (below branches) measured as number of substitutions per site. Numbers next to nodes indicate percent support from bootstrap analysis (1000 replicates). Cyan brackets with asterisks show significant differences in evolutionary rates compared to CYP734A7 (bold) as outgroup as assessed by Tajima's relative rate test (p<0.01). CYP734A7 causes BR-deficient phenotypes when overexpressed (*Ohnishi et al., 2006*). The orange branch shows evidence for relaxed constraint, but not for positive selection based on branch-site tests 1 and 2. The full tree is shown in *Figure 4—figure supplement 1*. Full results of branch-site tests 1 and 2 are shown in *Figure 4—source data 2*.

The following source data and figure supplements are available for figure 4:

**Source data 1.** Comparison of synonymous and non-synonymous substitution rates between *CYP734A50* and *CYP734A51* genes.

**Source data 2.** Full results of branch-site tests 1 and 2 for relaxed constraints or positive selection.

**Figure supplement 1.** Phylogenetic analysis of CYP734A-like proteins.

**Figure supplement 2.** Analysis of *CYP734A50* and *CYP734A51* introns.

(*Uyenoyama, 2005*); at the same time, purifying selection should maintain the function of the gene. Indeed, the four introns of *P. veris CYP734A50* were exceptionally long and contained multiple transposon-derived repetitive sequences (*Figure 4—figure supplement 2*). Comparing *P. veris* and *P. forbesii CYP734A50* gave a Ka/Ks ratio of 0.56, considerably higher than that of *CYP734A51* at 0.31. This difference was entirely due to a higher rate of non-synonymous substitutions for *CYP734A50* (p<0.01), while the rate of synonymous substitutions was indistinguishable (*Figure 4—source data 1*). Similarly, Tajima's relative rate test indicated that CYP734A50 had accumulated significantly more amino-acid exchanges than CYP734A51 since their duplication (*Figure 4*). We used branch-site tests 1 and 2 developed by *Zhang et al., 2005* on the subtree shown in *Figure 4* to determine whether this was likely due to relaxed constraints or to positive selection on *CYP734A50* genes. No evidence for positive selection was found for the *CYP734A50* branches or the branch separating the two *CYP734A50* genes from their *CYP734A51* paralogues based on likelihood ratio test 2 (*Figure 4*; *Figure 4—source data 2*). By contrast, relaxed constraints were supported for the *PfoCYP734A50* branch by test 1 (orange branch in *Figure 4*). While rigorously ruling out positive selection would likely require sequences of the paralogues from more *Primula* species, based on the above results we favour the explanation that the faster rate of molecular evolution of CYP734A50 proteins is due to a reduced efficacy of purifying selection as predicted by theory.

Thus, our combined evidence indicates that *CYP734A50* represents the *G* locus in the *Primula* heterostyly supergene, whose expression in S-morph styles causes brassinosteroid degradation, limiting cell expansion and style elongation (*Figure 1—figure supplement 1*). Evolutionarily, *CYP734A50* arose via a lineage-specific gene duplication and presumably gain of style-specific expression. Two main models have been proposed to explain the evolutionary origin of distyly. While they differ in the sequence of events, assuming either that morphological features evolved first (*Lloyd and Webb, 1992*) or that intra-morph incompatibility arose first (*Charlesworth and Charlesworth, 1979*), they both postulate that the first morphological change was the reduction of style length by a dominant mutation. The proposed duplication and gain of style-specific expression of *CYP734A50* in the ancestor of the Primulaceae likely represents this dominant style-shortening mutation. As such, *CYP734A50* is a rare example of an identified causal locus in a supergene underlying an adaptive polymorphism (*Schwander et al., 2014*) that opens the way for understanding the evolutionary origin of the *Primula S*-locus as an iconic case of a complex supergene.

## Materials and methods

### Plant material and growth conditions

The *P. veris* plants used for RNA-seq analysis were from a commercially grown line obtained from Jungpflanzen Braun (Kurzlipsdorf, Germany) and were cultivated in the field in the Botanical Garden at the University of Potsdam. *P. veris* plants for testing co-segregation of the *CYP734A50* presence/absence polymorphism with the *S*-locus genotype were from a semi-natural population in Park Sanssouci, Potsdam (*Nowak et al., 2015*).

*P. vulgaris* plants for sampling styles for brassinosteroid measurements and for BR treatment experiments were commercially grown hybrids bred by Gartenbau Ebbing-Lohaus, Heiden, Germany and provided by Gärtnerei Schultz, Potsdam-Sacrow. For the duration of the BR treatment, these were grown in a glasshouse at 18°C. Flowers of naturally occurring long homostyles of *P. vulgaris* and of neighbouring L- and S-morphs were sampled in Somerset, UK, near Wyke Champflower (grid reference: ST656339; http://www.gridreferencefinder.com/), Wanstrow (ST718405) and Sparkford (ST620285); and in the Chilterns, UK, near Great Missenden (SP897014). Both the commercially grown and naturally occurring *P. vulgaris* plants were used for testing co-segregation of the *CYP734A50* presence/absence polymorphism with the *S*-locus genotype. Samples for qRT-PCR analysis were collected near Great Missenden (long homostyles); as there were no S-morph individuals in this population, L- and S-morph control plants were collected near Oxford. The experiment shown in *Figure 2D* used material sampled in 2015. A replicate experiment using material sampled in 2016 confirmed this result. The long homostyle mutant of *P. vulgaris* carrying the exon-4 deletion was provided by Gartenbau Ebbing-Lohaus, Heiden, Germany, where it arose spontaneously as a single plant in a large *P. vulgaris* population; the exon-4 deletion mutant of *P. x pruhoniciana* (a hybrid of

*P. juliae* and *P. vulgaris*) was found in a local garden center in Berlin. Therefore, both mutants were considered to be independently arisen.

*P. forbesii* seeds were obtained from the Botanical Garden at the University of Zurich (accession 20050032) and grown in a growth room at 18°C, initially under short-day conditions (8 hr light, 16 hr dark) to stimulate flowering, then under long-day conditions (16 hr light, 8 hr dark). The long homo-style recombinants were found to segregate in this population.

Material for the remaining species (*P. elatior, P. juliae, P. grandis, P. frondosa, P. farinosa, P. scotica, P. halleri,* and *P. vulgaris*) was sampled either from living plants cultivated at the Botanical Garden of the University of Zurich, or from material collected in the field. *Primula elatior* was collected from several (semi) natural populations in Switzerland by BK: Botanical Garden of Zürich (ZH; 47°21′29′′N, 8°33′37′′E), Niederönz (BE; 47°11′41′′N, 7°41′37′′), Staufenbach (BE; 47°08′58′′N, 7°43′53′′E), and Allerheiligenberg (SO; 47°20′58′′, 7°48′50′′E). *Primula farinosa* was collected from a natural population at Alpe Campolungo (TI, Switzerland; 64°28′15′′N, 8°43′14′′E) by Jurriaan M. de Vos and seeds (accession IT0Z-20050747) were obtained from the Paradisia Alpine Botanical Garden (Cogne, Italy). *Primula frondosa* seeds (accession XX0Z-20070548) were obtained from the Botanical Garden Hamborn (Duisburg, Germany). *Primula grandis* seeds (accessions XX0Z-20110194 and XX0Z-20130917) were obtained from the Akureyri Botanical Garden (Akureyri, Island) and the Scottish Rock Garden Club, respectively. *Primula halleri* was collected from several natural populations across the Binntal (VS, Switzerland; 46°22′46′′N, 8°14′52′′E) by Jurriaan M. de Vos. *Primula juliae* was collected from a natural population in the Shzoma gorge (Lagodekhi, Georgia; 41°52′27.5′′N, 46°19′47.0′′E) by Rita Ganz. *Primula scotica* was collected from a natural population at Faraid Head (Sutherland, Scotland; 58°37′00.05′′N, 4°46′00.00′′W) by Jurriaan M. de Vos. *Primula vulgaris* was collected from two semi natural populations in Switzerland by BK: Botanical Garden of Zürich (ZH; 47°21′29′′N, 8°33′37′′E) and Niederönz (BE; 47°11′41′′N, 7°41′37′′E).

## RNA-seq

Styles and petal tubes with attached anthers of L- and S-morph flowers of *P. veris* were sampled when petals were between 4 and 10 mm long, corresponding to the time when the difference in style length and anther position develops (*Stirling, 1932*; *Webster and Gilmartin, 2006*). Three replicate samples were collected per morph and organ, pooling organs of 25 plants per sample. Similarly, styles of L- and S-morph flowers of *P. forbesii* were sampled from young, unopened flower buds. One sample per morph was collected. Tissue was immediately frozen in liquid nitrogen following manual dissection of the flowers. Total RNA was extracted using RNeasy Plant Mini Kit (Qiagen). TruSeq RNA libraries were generated according to the manufacturer's instructions (Illumina) and sequenced using the Illumina HiSeq2500 instrument (2 × 100 cycles), essentially as described (*Mascher et al., 2013*).

## *Primula forbesii* genome sequencing

Genomic DNA of S- and L-morph plants of *P. forbesii* was extractred using DNeasy Plant Mini Kit (Qiagen). For sequencing one NextSeq library each was generated per morph as described by the manufacturer (Illumina) and sequenced on a NextSeq500 instrument (paired-end mode, 2 × 150 cycles). In addition, a TruSeq DNA library was constructed for the S-morph as described by the manufacturer (Illumina); sequencing was performed using an Illumina HiSeq2500 instrument (paired-end mode, 2 × 100 bp reads), essentially as described (*Mascher et al., 2013*).

## Bioinformatic analysis

RNA-seq data were processed using Trimmomatic (*Bolger et al., 2014*) to remove adapter sequences. Quality control was done using FastQC (http://www.bioinformatics.bbsrc.ac.uk/projects/fastqc). Transcriptome assembly was done separately for *P. veris* and *P. forbesii* using Trinity version 2.1.1 (*Grabherr et al., 2011*). Assemblies were further explored using Trinity downstream analysis scripts to estimate abundances using Kallisto (*Bray et al., 2016*) and to obtain differential gene expressions for *P. veris* samples by comparing all samples to S-morph style ones using DESeq2 (*Love et al., 2014*). Genes with higher expression in S-morph styles for all three comparisons having a p-value below 0.05 were considered as *G* locus candidates. Functional annotations for those genes were based on tblastx search against TAIR10 cDNA sequences. *PveCYP734A50* exons were identified

based on the *P. veris* draft genome (*Nowak et al., 2015*). The corresponding *P. forbesii Pfo-CYP734A50* was identified by blastn and tblastx alignments. *PveCYP734A50* and *PfoCYP734A50* coding sequences were confirmed by Sanger sequencing and are included in *Supplementary file 2*. They are also available in GenBank under accession numbers KX589238, KX589239, KX589240 and KX589241. *PveCYP734A50* intron lengths were compared to genome-wide *P. veris* structural gene annotations (*Nowak et al., 2015*). Repeats and transposable elements in the *PveCYP734A50* locus were identified by comparison to RepBase (*Jurka et al., 2005*) version 21.03 using CENSOR (*Kohany et al., 2006*). The *CYP734A50* gene was named as such after consultation with Dr. David Nelson (http://drnelson.uthsc.edu/CytochromeP450.html).

*P. forbesii* S-morph DNA-seq data obtained using HiSeq2500 sequencing were assembled using SOAPdenovo2 version 2.04 (*Luo et al., 2012*) to identify *PfoCYP734A50* exons. Close *CYP734A50* homologs for both *P. veris* and *P. forbesii* were identified by tblastx alignments against the corresponding genome and transcriptome assemblies.

Public *P. vulgaris* DNA-seq data from NCBI SRA projcect PRJEB9683 for L- and S-morphs were mapped against the *PveCYP734A50* and *PveCYP734A51* exons separated by stretches of 100 Ns using bwa mem (*Li, 2013*). Our *P. forbesii* DNA-seq data was mapped the same way against the *Pfo-CYP734A50* and *PfoCYP734A51* exons. Mapped reads were counted to obtain a relative abundance estimate.

## Expression analysis

To determine gene expression in different floral organs, recently opened flowers of L- and S-morph plants were manually dissected and the tissue frozen immediately in liquid nitrogen. Total RNA was extracted using RNeasy Plant Mini Kit (Qiagen). Reverse transcription was performed using oligo(dT) priming and SuperScript III (Invitrogen). RT-PCR products were separated on agarose gels and visualized by ethidium bromide staining. Quantitative RT-PCR was performed using the Bioline SensiMix SYBR Low-ROX kit and a Roche LightCycler LC480. The *Primula TUBULIN* gene was used as a reference. Primers used for RT-PCR are indicated in *Supplementary file 1*.

## Phylogenetic analysis of *CYP734A50* homologues

CYP homologs were identified within the Phytozome version 11 (https://phytozome.jgi.doe.gov) by aligning CYP734A50 and CYP734A51 sequences from *P. veris* and *P. forbesii* using blastp. Multiple sequence alignments were generated using MUSCLE. Sequences used and multiple-sequence alignments are available as *Supplementary files 2–5*. The evolutionary history was inferred using the Neighbor-Joining method. The optimal tree with the sum of branch length = 14.64379199 is shown in *Figure 4—figure supplement 1*. The percentage of replicate trees in which the associated taxa clustered together in the bootstrap test (1000 replicates) are shown next to the branches. The tree is drawn to scale, with branch lengths in the same units as those of the evolutionary distances used to infer the phylogenetic tree. The evolutionary distances were computed using the Poisson correction method and are in the units of the number of amino acid substitutions per site. The rate variation among sites was modeled with a gamma distribution (shape parameter = 1). The analysis involved 109 amino acid sequences (*Supplementary files 2–5*). All positions containing gaps and missing data were eliminated. There were a total of 183 positions in the final dataset. Pair-wise Ka/Ks ratios were determined from MUSCLE-generated multiple-sequence alignments of coding sequences using the Nei-Gojobori model with uniform rates and complete deletion of missing data. Number of substitutions were extracted and the total number of sites calculated from the Ka and Ks values and the absolute numbers of substitutions; these were used in the Fisher's exact tests in *Figure 4—source data 1*. Qualitatively similar results were obtained when calculating Ka and Ks values using only pairwise deletions. Evolutionary rates were compared between CYP734A50 and CYP734A51 using Tajima's relative rate test (*Tajima, 1993*). Evolutionary analyses were conducted in MEGA7 (*Kumar et al., 2016*).

To test for selection we performed branch-site tests 1 and 2 as described by *Zhang et al. (2005)*, using codeml implemented in PAML (*Yang, 2007*) on the subtree shown in *Figure 4*. We performed a likelihood ratio test comparing model A with variable ω in the foreground branch to its null model with fixed ω value to identify positive selection. We did the same test comparing model A to model M1a to identify overall relaxed constraints or positive selection. Codon alignments were done using

TranslatorX (*Abascal et al., 2010*) in combination with MUSCLE (*Edgar, 2004*). Full results of the tests are shown in *Figure 4—source data 2*.

## PCR-based genotyping

Genomic DNA was extracted from leaf material using the DNeasy Plant Mini Kit (Qiagen). PCR-genotyping was performed using the primers indicated in *Supplementary file 1*. Primers for genotyping for presence/absence of *CYP734A50* across the different species in *Figure 2* were designed based on conservation between the *P. veris* and *P. forbesii CYP734A50* sequences. To monitor successful PCR amplification, internal-control primers targeting *ITS* were included in each PCR reaction. PCR products were separated on agarose gels and visualized by ethidium bromide or UView 6x Loading Dye (Bio-Rad Laboratories, Inc) staining.

## Brassinosteroid measurements and treatment

Styles were sampled from mixed-stage unopened flower buds of L- and S-morph *P. vulgaris* plants, ranging from a petal length of about 10 mm to buds just before opening. Tissue was frozen immediately in liquid nitrogen. After weighing, the tissue was freeze-dried. Extraction, semi-purification, and quantification of castasterone and brassinolide were carried out as described (*Sawai et al., 2014*). While castasterone was readily detectable in styles of L-morph plants, but virtually undetectable in styles of S-morph plants, brassinolide was undetectable in styles of either morph.

For the brassinosteroid treatment experiment, young flower buds of commercially grown *P. vulgaris* and of *P. forbesii* were injected every other day with solutions containing 1 nM, 100 nM or 10 µM homobrassinolide (dissolved in DMSO), 0.1% Tween-20 or a mock solution containing 0.1% DMSO (corresponding to the DMSO concentration in the solution with 10 µM homobrassinolide), 0.1% Tween. Injections were started on flower buds with petals of about 8 and 3 mm length, respectively. Once fully opened, flowers were bisected, photographed and organ sizes were determined using ImageJ. Styles were excised and placed in 1x PBS for processing for confocal microscopy.

To measure cell lengths in styles, styles were briefly dewaxed in chloroform and stained with propidium iodide. Stained tissue was imaged with a Zeiss LSM 710 using a 488 nm excitation laser line while emission was detected at 560–750 nm for FM4-64. Style-cell lengths were determined from digital micrographs using ImageJ.

## Virus-induced gene silencing

To prepare vectors for virus-induced gene silencing (VIGS), a fragment containing exon 4 and exon 5 of *CYP734A50* in *P. forbesii* was cloned and inserted into vector TRV2 (*Liu et al., 2002*) to produced TRV2-CYP734A50. As a control, a fragment of the *PHYTOENE DESATURASE (PDS)* gene was used. Primer sequences are provided in *Supplementary file 1*. Vectors TRV1 (*Liu et al., 2002*), TRV2-CYP734A50 and TRV2-PDS were then introduced into *Agrobacterium tumefaciens* strain GV3101. Preparation steps for VIGS assay were done as described (*Padmanabhan and Dinesh-Kumar, 2009*). *P. forbesii* plants were grown under long-day conditions (16 hr light, 8 hr dark) at 18°C. *Agrobacterium* cultures were infiltrated into inflorescence stems using a 2-ml syringe with needle. The treatment was repeated after one week. In total 22 plants were infected with *Agrobacterium*.

Flowers from treated plants showing rescued style length were photographed. Style length and anther height were measured using ImageJ. 6–8 styles were dissected for total RNA extraction and the expression analysis was performed as described above with primers in *Supplementary file 1*.

## Statistical analysis

No statistical methods were used to predetermine sample sizes; rather, the maximum number of available plants was used. The number of replicates performed is indicated in the respective figure legends. No data points were excluded. Trait values in the BL treatment and VIGS experiment departed from normality in some groups, based on a Shapiro-Wilk-test; also, variance were unequal between some of the groups. Student's *t*-test is robust to deviations from normality when sample sizes are large (n > 25) and comparable between the groups, both of which is the case here, and when using a two-sided test (*Sawilowsky and Blair, 1992*); also a correction for unequal variances is available. We therefore compared style lengths and anther heights of BL-treated to mock-treated groups using a two-sided t-test assuming unequal variances. Similarly, we compared style lengths

and anther heights of VIGS-treated S-morph plants with mock-treated S-morph plants, and untreated S-morph plants with mock-treated S-morph plants using a two-sided t-test assuming unequal variances. The Bonferroni method was used to correct for multiple testing.

## Acknowledgements

We are grateful to the staff of the Botanical Garden of the University of Potsdam for plant care and Jurriaan de Vos, Rita Ganz and Annette Becker for material. We thank John Richards for discussion and for suggesting the use of *Primula forbesii*. We thank Michael Hofreiter and Nils Stein for additional Illumina sequencing, and Ines Walde and Sandra Driesslein for help with library construction and sequencing. This work was supported by initial grants from the Swiss National Science Foundation (no. 3100-061674.00/1) and the Claraz Foundation to EC.

## Additional information

### Competing interests

TE-L: The owner of Ebbing-Lohaus Gartenbau and an employee of Ebbing-Lohaus Vertriebsgesellschaft mbH. The other authors declare that no competing interests exist.

### Funding

| Funder | Grant reference number | Author |
| --- | --- | --- |
| Schweizerischer Nationalfonds zur Förderung der Wissenschaftlichen Forschung | 3100-061674.00/1 | Elena Conti |
| Claraz Foundation | | Elena Conti |

The funders had no role in study design, data collection and interpretation, or the decision to submit the work for publication.

### Author contributions

CNH, CK, BK, Acquisition of data, Analysis and interpretation of data, Drafting or revising the article; AS, YT, AH, HS, LA, Acquisition of data, Analysis and interpretation of data; HB, IB, MB, Acquisition of data, Contributed unpublished essential data or reagents; MDN, Provided prepublication access to P. veris genome sequence, Contributed unpublished essential data or reagents; TE-L, Provided the long-homostylous P. vulgaris mutant, Contributed unpublished essential data or reagents; EC, Analysis and interpretation of data, Drafting or revising the article; ML, Conception and design, Acquisition of data, Analysis and interpretation of data, Drafting or revising the article

### Author ORCIDs

Christian Kappel, http://orcid.org/0000-0002-1450-1864
Isabel Bäurle, http://orcid.org/0000-0001-5633-8068
Michael Lenhard, http://orcid.org/0000-0001-8661-6911

## Additional files

### Supplementary files

• Supplementary file 1. Oligonucleotide sequences used.

• Supplementary file 2. Coding sequences of *CYP734A* family members from sequenced genomes in fasta format.

• Supplementary file 3. Multiple-sequence alignment of coding sequences of *CYP734A* family members in MEGA format.

• Supplementary file 4. Protein sequences of *CYP734A* family members from sequenced genomes in fasta format.

• Supplementary file 5. Multiple-sequence alignment of protein sequences of *CYP734A* family members in MEGA format.

### Major datasets

The following dataset was generated:

| Author(s) | Year | Dataset title | Dataset URL | Database, license, and accessibility information |
|---|---|---|---|---|
| Cuong Nguyen Huu, Christian Kappel, Michael Lenhard | 2016 | Primula S-locus | http://www.ncbi.nlm.nih.gov/bioproject/PRJNA317964 | Publicly available at the NCBI BioProject database (accession no: PRJNA317964) |

The following previously published dataset was used:

| Author(s) | Year | Dataset title | Dataset URL | Database, license, and accessibility information |
|---|---|---|---|---|
| The Genome Analysis Centre (Norwich UK) | 2015 | The genetical and molecular architecture of the Primula S locus supergene | http://www.ncbi.nlm.nih.gov/bioproject/PRJEB9683 | Publicly available at the NCBI BioProject database (accession no: PRJEB9683) |

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
