## [Decision Letter]

Thank you for submitting your article "Presence versus absence of *CYP734A50* underlies the style-length dimorphism in primroses" for consideration by *eLife*. Your article has been favorably evaluated by Detlef Weigel as the Senior editor and three reviewers, one of whom is a member of our Board of Reviewing Editors. Two of the three reviewers have agreed to reveal their identity: Elena Kramer (Reviewer #2) and John Pannell (Reviewer #3).

The reviewers have discussed the reviews with one another and the Reviewing Editor has drafted this decision to help you prepare a revised submission.

As you can see from their comments, all reviewers (and the Senior Editor) are very positive about your work and support publication. They have some suggestions for improvements, which should hopefully not too difficult to address. You can find those remarks in the reviews below.

*Reviewer #1:*

This manuscript by Lenhard and co-workers is an excellent piece of work that reports the identification of the causal variation within a supergene locus. Inherently, this is a difficult task because of the lack of recombination events within a supergene locus, but here the authors do a superb job in working around this. Basically, they were looking for presence-absence signs of gene expression in tissue-specific RNAseq experiments. This allowed them to home in on a candidate gene that codes for a brassinosteroid-degrading enzyme, which made sense because brassinosteroids are crucial for cell elongation and the heterostyly phenotypes of primula are due to differential cell elongation. Subsequently, they could show that the presence-absence of the pertinent *CYP734A50* gene holds up in S- and L-morphs across several primula species as well as mutant lines, and follow up physiological assays support a direct role of *CYP734A50* in the phenotype. Thus, despite the absence of a classic genetic linkage analysis, the identity of *CYP734A50* as the causal locus has solid support from cross-species/variety mapping. There are only a few, minor improvements that I would like to ask for:

Introduction: "Darwin's seminal book". Maybe add the book's title here, I doubt many colleagues are aware of it. Also, the doi in the corresponding reference is missing.

The S-morph haplotype is dominant, but only exists in heterozygous state in nature, why is that? Because of incompatibility, or because of a detrimental dosage phenotype of *CYP734A50* homozygosity?

Regarding the amplification of *CYP734A50* genes from different *Primula* species, was this initially done with degenerate primers? And then the genotyping was performed with specific primers based on the amplified fragments? I think the authors should provide a list of oligonucleotides used to amplify different fragments in different genotypes.

Regarding the brassinosteroid deficiency in S-morphs, why did the authors only determine the castasterone level? Why not brassinolide itself? Is castasterone the main active brassinosteroid in *Primula*? And is it the direct substrate for *CYP734A50*, or is it brassinolide? Please specify and amend accordingly.

*Reviewer #2:*

In this manuscript, the authors provide a very thorough analysis of the genetic basis for style length differences segregating among individuals of *Primula*. This phenomenon, termed distyly, has only evolved once in this genus, although it has also evolved independently in other taxa. The authors used differential expression to identify a candidate gene, *CYP734A50*, that is only expressed in S-morphs. This cytochrome P450 is involved in degrading brassinosteroids, a class of hormones often associated with cell elongation. The authors further demonstrate that 1) style-specific expression of the gene is associated with the S-morph phenotype of multiple *Primula* species, 2) the presence/absence of the locus specifically segregrates with the S-morph phenotype in multiple species, 3) suppression of activity lengthens styles, 4) presence of activity results in lower brassinosteroid concentrations, 5) application of excess brassinosteroids can rescue the short style phenotype, and 6) presence/absence of the locus correlates with cell length. The data are clearly presented and discussed. I think that this is a really lovely paper that addresses a classic question in plant genetics and pollination biology. I have no suggestions for major revisions.

*Reviewer #3:*

This manuscript reveals the identity of the gene responsible for the expression of short styles in distylous species of the primrose family. The gene, *CYP734A50*, degrades brassinosteroids, which promote cell elongation. Absence of this gene leads to short short-styled individuals; indeed, the gene was found only in short-styled individuals of several species of the genus *Primula*, and estimates of transcript read coverage were consistent with the gene being hemizygous in short-styled individuals. Also consistent with the inference that *CYP734A50* is indeed the gene responsible for style length in *Primula* is the fact that it was absent not only in typical long-styled individuals, but also in long homostyled individuals of *P. forbesii* or *P. grandis* (though interestingly not in those of *P. vulgaris*, suggesting that homostyly in the latter may have evolved via a different route). Virus-induced gene silencing of *CYP734A50* showed reduced expression of the gene in short-styled plants, and the development of longer styles. Moreover, the brassinosteroid castasterone, which is degraded by *CYP734A50*, was present in L-morph but not S-morph styles. I am not qualified to comment on many of the methodological details behind these aspects of the study, but the case made by the authors seems convincing on the whole, and I found the study exciting. It is also very well written.

The authors carry out an analysis of sequence variation at *CYP734A50* and find that its Ka/Ks ratio is twice as high as that found for its paralog *CYP734A51*. I would like to see the P-value for this analysis given in the text itself. The authors infer from this difference that there has been reduced purifying selection on *CYP734A50*, consistent with an hypothesis of its likely reduced effective population size. But how can they rule out the alternative hypothesis that the greater number of non-synonymous substitutions have in fact been driven by selection, e.g., in establishing the novel function of this gene in stylar tissue of the S-morph? It would be interesting to know when the substitutions actually occurred? Are they common to all species of *Primula*, i.e., did they occur at the base of the tree, or have different non-synonymous substitutions accumulated in different lineages during diversification of the genus? The latter possibility would perhaps be consistent with the authors' hypothesis of relaxed purifying selection, whereas the former would be more consistent with their fixation by selection. This would be well worth considering with more analytic rigour.

The main results of this paper are, I think, very exciting and represent an important advance. However, I feel the authors do not do their paper justice in their discussion of its evolutionary implications. It is commonly thought that distyly almost always evolves through the selection of a dominant mutation that suppresses stylar growth in the S-morph, and that, subsequently, anther lengths of the two morphs are adjusted, i.e., that the starting point is a population with approach herkogamy, and that the first step towards distyly is the spread of a dominant mutation that reduces style length. The authors' findings are entirely consistent with these theoretical explanations, yet they make too little of this fact. (Here, it would also be worth explaining briefly to the reader why the first mutation is more likely to be one that reduces rather than extends style length.) The authors might also stress the significance of the fact that the dominant action of *CYP734A50* is nicely explained by the gene duplication that their study reveals.

---

## [Author Response]

*As you can see from their comments, all reviewers (and the Senior Editor) are very positive about your work and support publication. They have some suggestions for improvements, which should hopefully not too difficult to address. You can find those remarks in the reviews below.*

*Reviewer #1:*

*This manuscript by Lenhard and co-workers is an excellent piece of work that reports the identification of the causal variation within a supergene locus. Inherently, this is a difficult task because of the lack of recombination events within a supergene locus, but here the authors do a superb job in working around this. Basically, they were looking for presence-absence signs of gene expression in tissue-specific RNAseq experiments. This allowed them to home in on a candidate gene that codes for a brassinosteroid-degrading enzyme, which made sense because brassinosteroids are crucial for cell elongation and the heterostyly phenotypes of primula are due to differential cell elongation. Subsequently, they could show that the presence-absence of the pertinent CYP734A50 gene holds up in S- and L-morphs across several primula species as well as mutant lines, and follow up physiological assays support a direct role of CYP734A50 in the phenotype. Thus, despite the absence of a classic genetic linkage analysis, the identity of CYP734A50 as the causal locus has solid support from cross-species/variety mapping. There are only a few, minor improvements that I would like to ask for:*

*Introduction: "Darwin's seminal book". Maybe add the book's title here, I doubt many colleagues are aware of it. Also, the doi in the corresponding reference is missing.*

We have added the book title. As we could not locate a doi for the book, we have provided a link to the website where a digital version is available.

*The S-morph haplotype is dominant, but only exists in heterozygous state in nature, why is that? Because of incompatibility, or because of a detrimental dosage phenotype of CYP734A50 homozygosity?*

The absence of S-morph haplotype homozygotes is likely due to a recessive-lethal mutation on this haplotype, as suggested by genetic analysis. We have added an explanation to the Introduction.

*Regarding the amplification of CYP734A50 genes from different Primula species, was this initially done with degenerate primers? And then the genotyping was performed with specific primers based on the amplified fragments? I think the authors should provide a list of oligonucleotides used to amplify different fragments in different genotypes.*

We did not use degenerate primers; rather, the primers for genotyping for presence/absence of *CYP734A50* across the different species in Figure 2 were designed based on high conservation between the *P. veris* and *P. forbesii CYP734A50* sequences in exon 3. We have added more detailed information about which primers were used on which sequences to the list of oligonucleotides in [Supplementary-material SD5-data].

*Regarding the brassinosteroid deficiency in S-morphs, why did the authors only determine the castasterone level? Why not brassinolide itself? Is castasterone the main active brassinosteroid in Primula? And is it the direct substrate for CYP734A50, or is it brassinolide? Please specify and amend accordingly.*

We also attempted to measure brassinolide levels; however, it was undetectable in both morphs. We have added this clarification to the Results section. Unfortunately, we do not know what is the direct substrate for *CYP734A50*.

*Reviewer #3:*

*The authors carry out an analysis of sequence variation at CYP734A50 and find that its Ka/Ks ratio is twice as high as that found for its paralog CYP734A51. I would like to see the P-value for this analysis given in the text itself. The authors infer from this difference that there has been reduced purifying selection on CYP734A50, consistent with an hypothesis of its likely reduced effective population size. But how can they rule out the alternative hypothesis that the greater number of non-synonymous substitutions have in fact been driven by selection, e.g., in establishing the novel function of this gene in stylar tissue of the S-morph? It would be interesting to know when the substitutions actually occurred? Are they common to all species of Primula, i.e., did they occur at the base of the tree, or have different non-synonymous substitutions accumulated in different lineages during diversification of the genus? The latter possibility would perhaps be consistent with the authors' hypothesis of relaxed purifying selection, whereas the former would be more consistent with their fixation by selection. This would be well worth considering with more analytic rigour.*

To test for significance of the different Ka/Ks ratios, we determined Ka and Ks values for the pair-wise comparisons of *CYP734A50* and *CYP734A51* genes, determined the absolute number of substitutions and calculated the total number of synonymous and non-synonymous sites from these. Note that to do so, we could not use the Jukes-Cantor correction, as we had done for the previous version of the manuscript, which explains the slightly different Ka/Ks values given in the text compared to the previous version. Fisher’s exact test was used to test whether there was a difference in the number of non-synonymous substitutions or of synonymous substitutions between the *CYP734A50* and the *CYP734A51* paralogues. This indicated a significantly higher rate of non-synonymous substitutions in *CYP734A50*, but no difference for the synonymous substitutions. This conclusion was robust to different treatments of missing data (complete versus pair-wise deletion). The results of this calculation are shown in Figure 4—figure supplement 3.

To address the issue of relaxed purifying versus positive selection more rigorously, we have performed the branch-site tests developed by Zhang et al. as implemented in PAML. While test 1 can detect signatures of either relaxed or positive selection, but cannot discriminate between them, test 2 specifically asks for positive selection. The branch for *PfoCYP734A50* shows a significant signal in test 1, but not in test 2. These results are highlighted in Figure 4, and the full test results are included as Figure 4—figure supplement 4. There is no evidence for positive selection acting on the *CYP734A50* copies. Therefore, these results indicate to us that the faster rate of protein evolution of the CYP734A50 copies relative to the CYP734A51 paralogues is most likely due to less efficient purifying selection, as predicted by theory.

However, we agree that we likely lack the sampling depth across the *Primula* phylogeny to fully rule out positive selection.

*The main results of this paper are, I think, very exciting and represent an important advance. However, I feel the authors do not do their paper justice in their discussion of its evolutionary implications. It is commonly thought that distyly almost always evolves through the selection of a dominant mutation that suppresses stylar growth in the S-morph, and that, subsequently, anther lengths of the two morphs are adjusted, i.e., that the starting point is a population with approach herkogamy, and that the first step towards distyly is the spread of a dominant mutation that reduces style length. The authors' findings are entirely consistent with these theoretical explanations, yet they make too little of this fact. (Here, it would also be worth explaining briefly to the reader why the first mutation is more likely to be one that reduces rather than extends style length.) The authors might also stress the significance of the fact that the dominant action of CYP734A50 is nicely explained by the gene duplication that their study reveals.*

We have added a brief discussion of the two competing models for the origin of distyly and pointed out that the proposed duplication and gain of style-specific expression of *CYP734A50* in the ancestor of *Primulaceae* is likely to represent the dominant style-shortening mutation assumed as the first morphological change in both models.